



# Examining the operational use of avalanche problems with decision trees and model-generated weather and snowpack variables

Simon Horton, Moses Towell, and Pascal Haegeli

Simon Fraser University, Burnaby, BC, Canada

**Correspondence:** Simon Horton (shorton@avalanche.ca)

**Abstract.** Avalanche problems are used in avalanche forecasting to describe snowpack, weather, and terrain factors that require distinct risk management techniques. Although they have become an effective tool for assessing and communicating avalanche hazard, their definitions leave room for interpretation and inconsistencies. This study uses conditional inference trees to explore the application of avalanche problems over eight winters in Glacier National Park, Canada. The influence of weather and snowpack variables on each avalanche problem type were explored by analyzing a continuous set of weather and snowpack variables produced with a numerical weather prediction model and a physical snow cover model. The decision trees suggest forecasters' assessments are not only based on a physical analysis of weather and snowpack conditions, but also contextual information about the time of season, location, and interactions with other avalanche problems. The decision trees show clearer patterns when new avalanche problems were added to hazard assessments compared to when problems were removed. Despite discrepancies between modelled variables and field observations, the model-generated variables produced intuitive explanations for conditions influencing most avalanche problem types. For example, 72 h snowfall was the most significant variable for storm slab avalanche problems, skier penetration depth was the most significant variable for dry loose avalanche problems, and slab density was the most significant variable for persistent slab avalanche problems. The explanations for wind slab and cornice avalanche problems were less intuitive, suggesting potential inconsistencies in their application as well as shortcomings of the model-generated data. The decision trees illustrate how forecasters apply avalanche problems and can inform discussions about improved operational practices and the development of data-driven decision aids.

## 1 Introduction

Avalanche problems have become a fundamental tool for assessing and communicating avalanche hazard. Avalanche problems are defined as a "set of factors that describe the avalanche hazard", where each problem describes a set of snowpack, weather, and terrain factors that require distinct risk management techniques (Statham et al., 2018a). For example, the risk management techniques for a storm slab avalanche problem caused by freshly deposited snow are distinct from a wet loose avalanche problem caused by cohesionless wet snow. Although different sets of problems are used in North America and Europe (European Avalanche Warning Services, 2017; Statham et al., 2018a), the concept has been widely adopted into professional avalanche risk management and backcountry recreation.



In North America, avalanche problems are applied using a standardized workflow known as the conceptual model of avalanche hazard (Statham et al., 2018a). Problems are assessed by answering four questions: what type of problem exists, where are they located in terrain, how likely are avalanches, and how big are the expected avalanches. Avalanche forecasters use expert judgment to answer these questions based on their synthesis of weather, snowpack, and avalanche observations over time (LaChapelle, 1980). By providing a structured workflow, the conceptual model of avalanche hazard intends to streamline this process and avoid issues in human judgment and decision-making. However, recent evidence has shown inconsistencies remain between individual forecasters and different organizations when it comes to applying avalanche danger ratings and avalanche problems (Clark, 2019; Lazar et al., 2016; Statham et al., 2018b; Techel et al., 2018). Guidelines for applying avalanche problems have been suggested (Klassen, 2014; Lazar et al., 2012; Müller et al., 2018; Müller et al., 2016), but in practice the diversity of data sources, forecasting contexts, and individual interpretations make them difficult to apply consistently.

Data-driven decision aids are a potential tool for improving the accuracy and consistency of hazard assessments (Canadian Avalanche Association, 2016b). Decision aids can include both statistical models that interpret available data and physical models that generate new data that would otherwise not be available from field observations. Statistical approaches to avalanche forecasting have been developed using weather and avalanche observations for a long time. Popular methods include nearest-neighbour analysis (Brabec and Meister, 2001; Obled and Good, 1980; Zeidler and Jamieson, 2004), discriminant analysis (Floyer and McClung, 2003; McClung and Tweedy, 1994), support vector machines (Pozdnoukhov et al., 2011), and classification trees (Blattenberger and Fowles, 2016; Yokley et al., 2014). These statistical models typically predict avalanche activity or an avalanche danger level. Since snowpack conditions are more difficult to characterize with numeric variables, including snowpack data in statistical models has typically relied on simple variables describing the snowpack (e.g. snow height) or with expert systems that emulate human decision making processes (Giraud, 1992; Schweizer and Föhn, 1996).

Physical models offer additional data to supplement field observations, most notably weather data from numerical weather prediction models and snowpack data from snow cover models. Weather models have been shown to add value to avalanche forecasts (Roeger et al., 2001; Schirmer and Jamieson, 2015), especially because of their capability to make predictions about future conditions. Snowpack models such as SNOWPACK and CROCUS can provide detailed simulations of snowpack structure, but their complex output is difficult to interpret (Morin et al., 2020). Giraud (1992) developed an expert-based system MEPRA to interpret the output of snowpack models with prescribed decision rules that emulate forecaster decision making. Following the tradition of the statistical models described above, Schirmer et al. (2009) explored a variety of machine learning techniques (i.e., classification trees, neural networks, support vector machines, and nearest-neighbour analysis) to statistically link the output of snowpack models to regional avalanche danger. Bellaire and Jamieson (2013) used classification trees to predict avalanche danger using four variables from simulated snow profiles – new snow over the past 24 and 72 h, the skier stability index, and critical layer depth.

While existing statistical studies have provided insights into the conditions influencing avalanche hazard, they have had limited adoption in operational practices. We attribute this to two main barriers. First, the relatively moderate predictive performance of these models as discussed by Schirmer et al. (2009), and second the fact that their predictions of avalanche danger level or an avalanche activity index only provide limited direct guidance on risk treatment measures. Given the broad



adoption of avalanche problems for describing avalanche hazard among practitioners, developing data-driven decision aids that support the assessment of avalanche problems could be a valuable next step. Haladuick (2014) developed separate classification trees for avalanche danger based on each avalanche problem type, showing distinct weather, snowpack, and avalanche variables that were relevant for each problem. Similarly, Müller et al. (2018) proposed automated approaches to assigning avalanche

problems based on available weather and snowpack data. Designing decision aids around avalanche problems in ways that emulate the hazard assessment of human forecasters holds promise to produce results that are more accessible, relevant, and easier to interpret.

The objective of this study is to contribute to this body of research by examining how avalanche problems are applied by avalanche forecasters. We use decision trees to explore links between avalanche problems identified in hazard assessments with

weather and snowpack variables generated by physical models. Numerical weather prediction and snow cover models provide a continuous and consistent stream of data that represents the type of information considered by forecasters in a way that facilitates statistical analyses that would not be possible with irregular field observations. The decision trees provide insights into how forecasters apply different avalanche problem types and show the potential benefits and limitations of incorporating data-driven decision aids into hazard assessments.

## 15  2   Methods

### 2.1   Study area

The study covers eight winter seasons at Glacier National Park, Canada, where Parks Canada produces daily avalanche forecasts to inform backcountry recreationists and protect the highway and railway that cross through the park (Fig. 1). The region is characterized by a transitional snow climate that favours heavy snowfall and the formation of multiple persistent weak layers

each season (Haegeli and McClung, 2007; Shandro and Haegeli, 2018). The park is a relatively small forecast region (1354 km$^2$) with a high density of field observations, resulting in avalanche forecasts that are produced with higher confidence than in larger Canadian forecast regions. Our study focuses on winter months when dry snow avalanches were the primary concern (i.e. December, January, February, and March), resulting in a study period covering 980 days over the 2012-13 to 2019-20 winter seasons.

### 25  2.2   Avalanche hazard assessments

Avalanche hazard assessments for Glacier National Park were compiled for each day of the study period (Table 1). Forecasters assessed avalanche problems at approximately 08:00 each morning to describe the current avalanche hazard conditions (i.e., nowcast) as part of the conceptual model of avalanche hazard workflow (Statham et al., 2018a). Our study focuses on the presence or absence of each problem type in the morning hazard assessment at each of the three elevation bands in the park

(i.e. below treeline, treeline, and alpine). Problem types were grouped into surface problem types (storm slab, wind slab, dry loose avalanche, wet loose avalanche, and cornice) and persistent problem types (persistent slab and deep persistent slab)

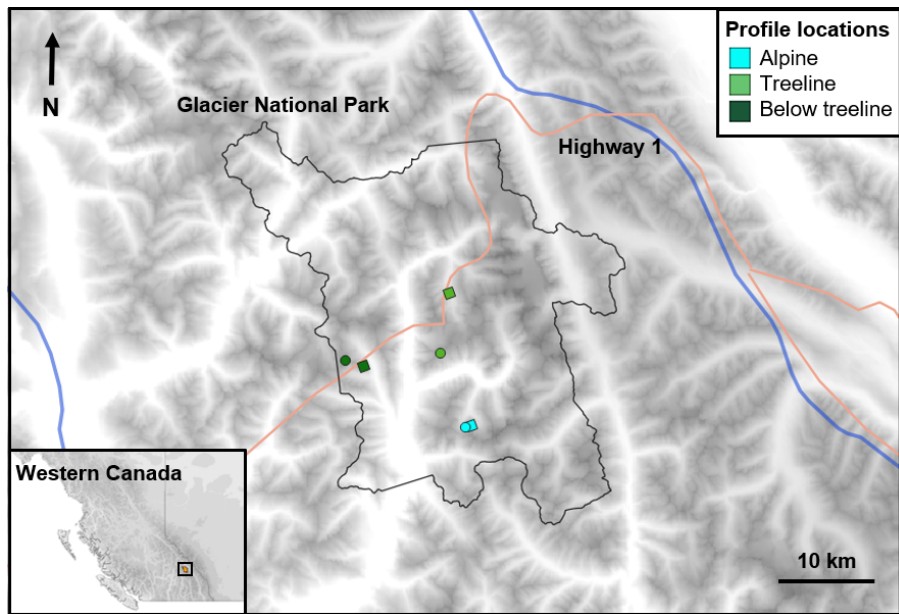

**Figure 1.** Location of Glacier National Park in western Canada and locations of grid points for each elevation band where weather and snowpack data were generated with numerical weather prediction and physical snow cover models. Grid points used before the 2017 model update are shown with circles and grid points used after the update are shown with squares.

based on the type of information needed for their assessment (Table 1). Over the study period the three most common avalanche problem types were persistent slab (present on 61% of the days), wind slab (50%), and storm slab avalanche problems (46%). Wet slab and glide slab avalanche problems were omitted from the analysis because they were never present during the study period.

To provide a meaningful analysis of persistent problem types, information about specific weak layers was also extracted from the hazard assessments. A list of relevant weak layers was produced for each season by reading the problem description and snowpack summary sections of the public avalanche forecast. This method identified 54 distinct weak layers that were associated with persistent problem types over the study period. Each weak layer was identified by a burial date and assigned a status that changed over time based on its role in persistent problem types. Once buried by new snow, layers were initially

assigned a status of *surface* in which case they may have been associated with a surface avalanche problem type (e.g., storm slab avalanche problem). The status changed to *active* when layers became attributed to a persistent avalanche problem type and then changed to *dormant* after they were no longer attributed to a persistent problem (either the problem was removed from the forecast or the problem became attributed to a different weak layer). Fifteen of the weak layers were attributed to a persistent problem at least one more time after a period of dormancy. However, since the reawakening of persistent weak layers is likely

driven by a different set of factors than the initial activation, we only considered the initial period of a layer contributing to a persistent slab problem in our analysis. Another 41 potential weak layers were described in the hazard assessments but never



**Table 1.** Avalanche problem types and their prevalence in Glacier National Park between 1 December and 31 March for the 2012-13 to 2019-20 winters.

| Avalanche problem group | Avalanche problem type | Description (edited from Statham et al., 2018a) | Percentage of days with problem | Percentage of days with problem by elevation band (below treeline/treeline/alpine) |
|---|---|---|---|---|
| Surface | Storm slab (STORM) | Cohesive slab of soft new snow (also called a direct-action avalanche) | 46 | 32/44/45 |
| | Wind slab (WIND) | Cohesive slab of locally deep, wind deposited snow | 50 | 3/40/50 |
| | Dry loose avalanche (DRYL) | Cohesionless dry snow starting from a point (also called a sluff or point release) | 19 | 13/17/16 |
| | Wet loose avalanche (WETL) | Cohesionless wet snow starting from a point (also called a sluff or point release) | 10 | 9/8/5 |
| | Cornice (CORN) | Overhanging mass of dense, wind deposited snow jutting out over a drop off in the terrain | 9 | 0/1/9 |
| Persistent | Persistent slab | Cohesive slab of old and/or new snow that is poorly bonded to a persistent weak layer and does not strengthen or strengthens slowly over time | 61 | 35/60/53 |
| | Deep persistent slab | Thick hard cohesive slab of old snow overlying an early-season persistent weak layer located in the lower snowpack or near the ground. Typically characterized by low likelihood and large destructive size | 3 | 1/3/3 |

became attributed to persistent problems. These layers were also documented to compare with the layers that were attributed to persistent problems.


## 2.3 Weather and snowpack data

Weather and snowpack variables were produced for each day of the study period using data from a numerical weather prediction model and a physical snow cover model. Variables were selected to be similar to the types of information forecasters routinely consider in their hazard assessments. Model-generated data have a decisive advantage for our study as their continuity and

consistency facilitate statistical analyses that would not be possible with the irregular field observations commonly used by avalanche forecasters.

### 2.3.1 Weather variables

Weather variables were compiled for each day of the study period using hourly data from a numerical weather prediction model. Data was compiled from the High Resolution Deterministic Prediction System (HRDPS) produced by the Meteorological

Service of Canada (Milbrandt et al., 2016). The HRDPS has 2.5 km grid spacing with initial and boundary conditions produced by a coarser regional scale deterministic model.

A single representative grid point was chosen for each elevation band in Glacier National Park (Fig. 1). Preliminary analyses revealed that the gridded weather data was relatively homogenous across the park, which is mainly due to the 2.5 km horizontal resolution. This means the selection of grid points should not have a strong influence on our analysis. The main source of

variability amongst the 236 grid points in the park was their elevation. Accordingly, grid points were chosen primarily based on their elevation with a secondary consideration being locations where the forecasted precipitation amounts were average relative to all grid points. It is important to note that the HRDPS underwent two major changes over the study period; an upgrade prior to the 2015 winter that improved the overall quality of the forecasts and a switch to a new grid prior to the 2018 winter that required a new set of grid points for subsequent winters (Faucher, 2016).

To account for the evolutionary nature of avalanche conditions and provide a meaningful summary of the weather conditions contributing to avalanche problems, hourly weather data were aggregated into summary statistics covering the 24 and 72 h periods prior to the hazard assessment (Table 2). Previous studies have consistently found weather conditions over these time periods as strong predictors of avalanche hazard (e.g. Schirmer et al., 2009; Yokley et al., 2014). Hourly surface weather data was extracted from the most up-to-date HRDPS forecasts, which was reinitialized every six hours. Precipitation was

accumulated over the previous 24 and 72 h periods and partitioned into solid, liquid, and total components (using an air temperature threshold of +0.5 °C). Minimum, maximum, and average air temperature (2 m above surface) and hourly wind speed (10 m above surface) were calculated over the same time periods. Accumulated incoming shortwave and longwave radiation were calculated over the 24 and 72 h periods, as well as the maximum intensity of incoming shortwave radiation.

### 2.3.2 Snowpack variables

Snowpack conditions were simulated with the SNOWPACK snow cover model (Lehning et al., 1999). The snowpack structure at each elevation band in Glacier National Park was simulated using the hourly weather data extracted from the representative HRDPS grid points (Fig. 1). Default SNOWPACK settings were used with wind transport disabled to produce flat field snow





**Table 2.** Weather variables calculated from hourly data from the HRDPS numerical weather prediction model.

| Variable name | Description |
| --- | --- |
| QPF SNOW/RAIN/TOTAL | Precipitation over last 24/72 h partitioned into solid/liquid/total amounts (mm) |
| TA MIN/MAX/AVG | Minimum, maximum, and average hourly temperature over last 24/72 h (° C) |
| VW MIN/MAX/AVG | Minimum, maximum and average hourly wind speed over last 24/72 h (m s$^{-2}$) |
| ISWR MAX/SUM | Maximum (W m$^{-2}$) and total (MJ m$^{-2}$) incoming shortwave radiation over last 24/72 h |
| ILWR SUM | Total incoming longwave radiation over last 24/72 h (MJ m$^{-2}$) |

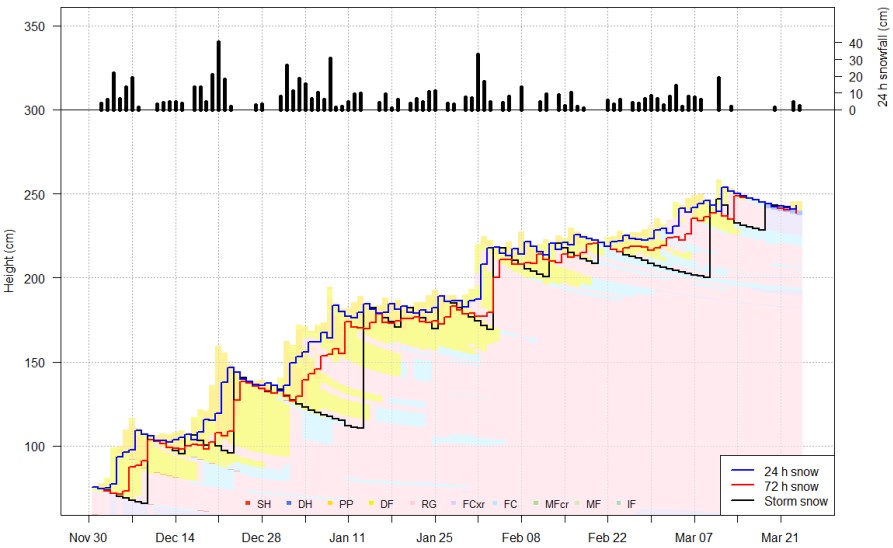

**Figure 2.** Daily snowfall amounts and snowpack interfaces identified in the upper snowpack for the 2019-20 treeline profiles. Interfaces include all snow deposited in the past 24 h, 72 h, and since the last day without snow. Bars show the modelled height of new snow each 24 h.

profiles representative of conditions in sheltered terrain. Daily snow profiles were produced at 08:00 to align with the hazard assessments.

To mimic existing forecaster practices, three interfaces were identified in the upper snowpack to produce variables relevant to surface avalanche problem types: a 24 h interface, a 72 h interface, and a storm interface (Fig. 2). The 24 and 72 h interfaces

5 were identified by searching for all snow deposited in the past 24 and 72 h. The storm interface was identified by selecting all snow deposited since the last full day without snow, similar to how accumulated snowfall is measured in the field on storm boards that are cleared in between stormy periods (Canadian Avalanche Association, 2016a).

Interfaces corresponding to the weak layers described in the hazard assessments were also identified to produce variables relevant to persistent avalanche problem types (Fig. 3). Since discrepancies arise between the burial date of layers used by

10 forecasters and the deposition dates used by SNOWPACK, a method to search for the most likely weak layer was developed.

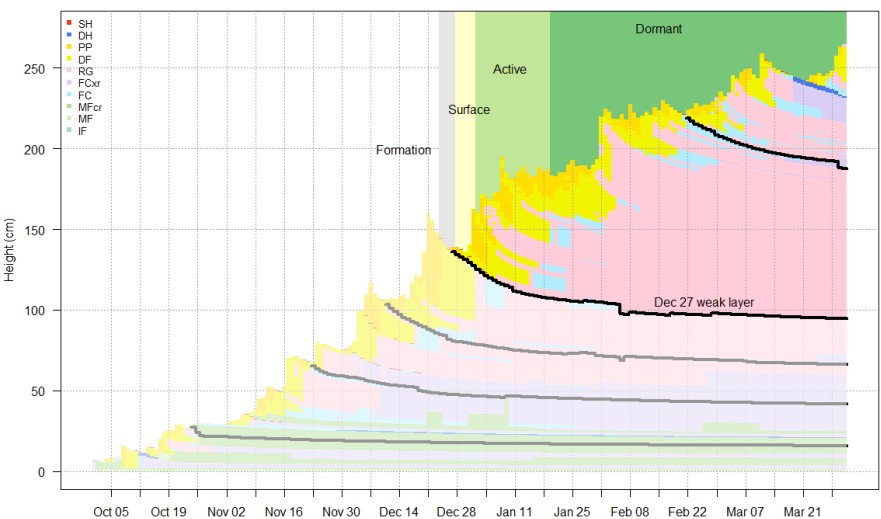

**Figure 3.** Weak layers identified in the 2019-20 treeline profiles. Five weak layers were listed in the hazard assessments, and the height of modelled layers most likely associated with these weak layers are shown with black lines. The status of the 27 December weak layer over time is highlighted. The layer was attributed to a persistent slab avalanche problem between 2 and 19 January and thus assigned the status *active*. The layer was assigned the status *surface* prior to this period and *dormant* after this period.

First, a range of possible deposition dates were identified for each weak layer by searching for clear weather days prior to the burial date. Once buried, weak layers were identified by searching for the weakest layer that fell within the range of potential deposition dates using SNOWPACK's structural stability index to identify the weakest layer (Schweizer et al., 2006).

Snowpack variables were extracted from the simulated profiles to summarize slab and weak layer properties for each of the identified interfaces (Table 3). Variables characterizing slabs included slab thickness, average and maximum density, average and maximum temperature, average and maximum hardness, and total liquid water content. Slab averaged values were calculated using a product sum with layer thickness. Variables characterizing weak layers included grain type, grain size, density, temperature, liquid water content, hardness, age (days since deposition), and whether there was a melt-freeze crust less than 5 cm below the weak layer. Four generic snowpack variables that did not depend on any interfaces were also calculated: height of snowpack, skier penetration depth, snow surface temperature, and the size of any surface hoar on the surface.

### 2.3.3 Verification of modelled data

Modelled snow heights and daily snowfall amounts were compared to assess the overall agreement between modelled variables and field observations. While an in-depth model validation was outside the scope of this study, sensitivity studies have shown physical snowpack models are most sensitive to precipitation inputs (Raleigh et al., 2015; Richter et al., 2020). Thus, the accuracy of modelled snow heights provides insights into the accuracy and reliability of the modelled data used in this study. Parks Canada manually observes snow height and daily snowfall amounts every day at their below treeline study plot





**Table 3.** Snowpack variables calculated from simulated snow profiles.

| Variable type | Variable name | Description |
| --- | --- | --- |
| Generic | HS | Height of snowpack (cm) |
| | SKI PEN | Ski penetration depth (cm) |
| | TSS | Snow surface temperature ($^\circ$C) |
| | HOAR | Grain size of any surface hoar on the surface (mm) |
| Slab | HN24/HN72/HST | Height of snow above 24 h/72 h/storm interfaces (cm) |
| | SLAB DENSITY MAX/AVG | Maximum and average density of slab (kg m$^{-3}$) |
| | SLAB TEMP MAX/AVG | Maximum and average temperature of slab ($^\circ$C) |
| | SLAB LWC | Total liquid water content of slab (kg m$^{-2}$) |
| | SLAB HARDNESS MAX/AVG | Maximum and average slab hardness |
| Weak layer | WL AGE | Weak layer age (days since deposition) |
| | WL DENSITY | Weak layer density (kg m$^{-3}$) |
| | WL TEMP | Weak layer temperature ($^\circ$C) |
| | WL LWC | Weak layer liquid water content (% volume) |
| | WL GRAIN SIZE | Weak layer grain size (mm) |
| | WL GRAIN TYPE | Weak layer grain type |
| | WL HARDNESS | Weak layer hardness |
| | WL ABOVE CRUST | Crust exists below weak layer (true/false) |

(Rogers Pass at 1315 m) and several times per week at their treeline study plot (Fidelity at 1905 m). These observations were compared to the corresponding modelled snow heights at the below treeline and treeline grid points using Spearman correlation coefficients, percent differences, hit rates, and false alarm rates.

## 2.4 Decision tree analysis

5 Decision trees are a class of machine learning methods that provide simple and interpretable visualizations of complex non-linear relationships. Unlike other machine learning methods that focus on predictive performance (e.g., neutral networks, random forests), decision trees present relationships in ways that more closely resemble the human decision-making processes and are thus a helpful tool to understand the avalanche problem identification and assessment process. While the classification and regression tree (CART) method by Breiman et al. (1984) has been widely used, the method has issues with overfitting

10 data and producing biased splits when the dependent variable is unevenly distributed. An alternate method that overcomes these weaknesses is conditional inference trees (Hothorn et al., 2006), a method that uses statistical hypothesis testing to recursively split data sets until no more statistically significant splits exist in the data. When visualized in a tree diagram, the most significant splits in independent variables are presented higher on the tree, and the terminal nodes at the bottom of the tree show the resulting distribution of the dependent variable.





**Table 4.** Contextual variables describing the time, location, and other problems in the hazard assessment.

| Variable type | Variable name | Description |
|---|---|---|
| Time | SEASON | Individual season |
| | SEASON ORDER | Before or after a specific season |
| | SEASON DAY | Day of the season (starting 1 October) |
| Location | ELEV BAND | Elevation band |
| Problems | STATUS STORM/WIND/DRYL/WETL/CORN | Problem status the same day (present or absent) |
| | STATUS STORM/WIND/DRYL/WETL/CORN PREV | Problem status the previous day (present or absent) |

In the present study, we employed conditional inference trees to explore relationships between weather variables, snowpack variables, and avalanche problem types using the *partykit* implementation in R (Hothorn and Zeileis, 2015). Since the focus of the analysis was to explore relationships rather than construct predictive models, the trees were made to be relatively simple by only including variable splits that had p-values smaller than 0.0001, except in analyses with smaller dataset where we kept

the threshold p-value at 0.05. To compare various decision tree models, contingency tables were produced by comparing the original observations with the classifications produced by each decision tree model (using the most common value in each terminal node). The accuracy, hit rate, and false alarm rate were calculated from the contingency table to describe the ability of each decision tree model to represent the original data (see Appendix A for definitions of these statistics).

### 2.4.1 Surface problem types

Decision trees were computed to determine the presence or absence of each surface avalanche problem type. The presence or absence of a surface problem at each elevation band for each day of the study period was fit to the relevant weather and snowpack variables for that elevation band and day (960 days which are equivalent to 2880 elevation band days). Weather variables included those listed in Table 2 for the past 24 and 72 h periods. Snowpack variables included the four generic snowpack variables in Table 3 and the slab and weak layer variables for the three types of surface interfaces (24 h, 72 h, and

since the last day without snow).

To offer a better understanding of how avalanche problems were applied, two decision trees were computed for each surface problem type: one using only weather and snowpack variables and a second using an additional set of contextual variables (Table 4). The contextual variables provided information about the hazard assessment and included temporal information (e.g. specific seasons, number of days since 1 October), elevation band, and the presence or absence of other surface problems

the same day and the previous day. This information further strengthened the representation of the evolutionary character of avalanche hazard in our analysis.





### 2.4.2 Persistent problems

Instead of analyzing whether persistent slab problems existed on a specific day, decision trees were computed to explore how persistent problem types related to specific weak layers. This approach reflects the more complex process of assessing persistent problems, where forecasters track the evolution of multiple weak layers over time rather than simply considering

the upper snowpack. For simplicity the analysis was limited to the treeline elevation band where persistent slab problems were most prevalent (Table 1). The analysis included the same weather variables as the surface problem analysis, but the snowpack variables were restricted to the slab and weak layer variables pertinent to the specific weak layer being tracked, plus the four generic snowpack variables shown in Table 3.

Several decision trees models were fit to compare conditions when weak layers were associated with persistent problems

(*active* status) with conditions before (*surface* status) and after (*dormant* status). First, a large decision tree with a three level dependent variable was used to examine the statuses of all 54 weak layers in a dataset that included 714 cases (i.e., weak layer days) of *surface* conditions, 866 cases of *active* conditions, and 280 cases of *dormant* conditions. The dormant cases were restricted to the first seven days after layers became dormant to focus on times when the problem was most relevant and avoid skewing the dataset with large numbers of cases with deeply buried dormant weak layers. Additional decision trees with binary

dependent variables were fit to focus on the onset and the end of persistent problem types in more detail. Onset was analyzed by narrowing the focus to the final three days of layers having a *surface* status and the first three days of having an *active* status. A similar decision tree was fit to focus on the transition from *active* to *dormant*, however the number of days before and after the transition was expanded to 14 since no significant variables were found for shorter time periods.

### 2.4.3 Variable selection

The weather and snowpack variables included in our dataset exhibit natural correlations, which can negatively affect statistical analyses and make the interpretation of decision trees more challenging. To avoid the problem, highly correlated weather and snowpack variables were removed from the analysis. Variables with pair-wise Spearman rank correlations above 0.9 were identified and the variable with the largest mean absolute correlation in the dataset was removed. A total of 23 highly correlated variables were removed from the surface problem dataset and 15 highly correlated variables were removed from the persistent

problem dataset (see Appendix B for correlation plots).

## 3 Results

### 3.1 Snow height verification

Modelled snow heights and daily snowfall amounts were strongly correlated with field observations over the study period in Glacier National Park (Table 5). Whereas snow heights were consistently underestimated at the treeline grid points (Fig. 4),

they were either overestimated, underestimated, or closely agreed at the below treeline grid points depending on the season. Over the entire study period the modelled snow heights were an average of 36% and 5% lower than the observed snow heights

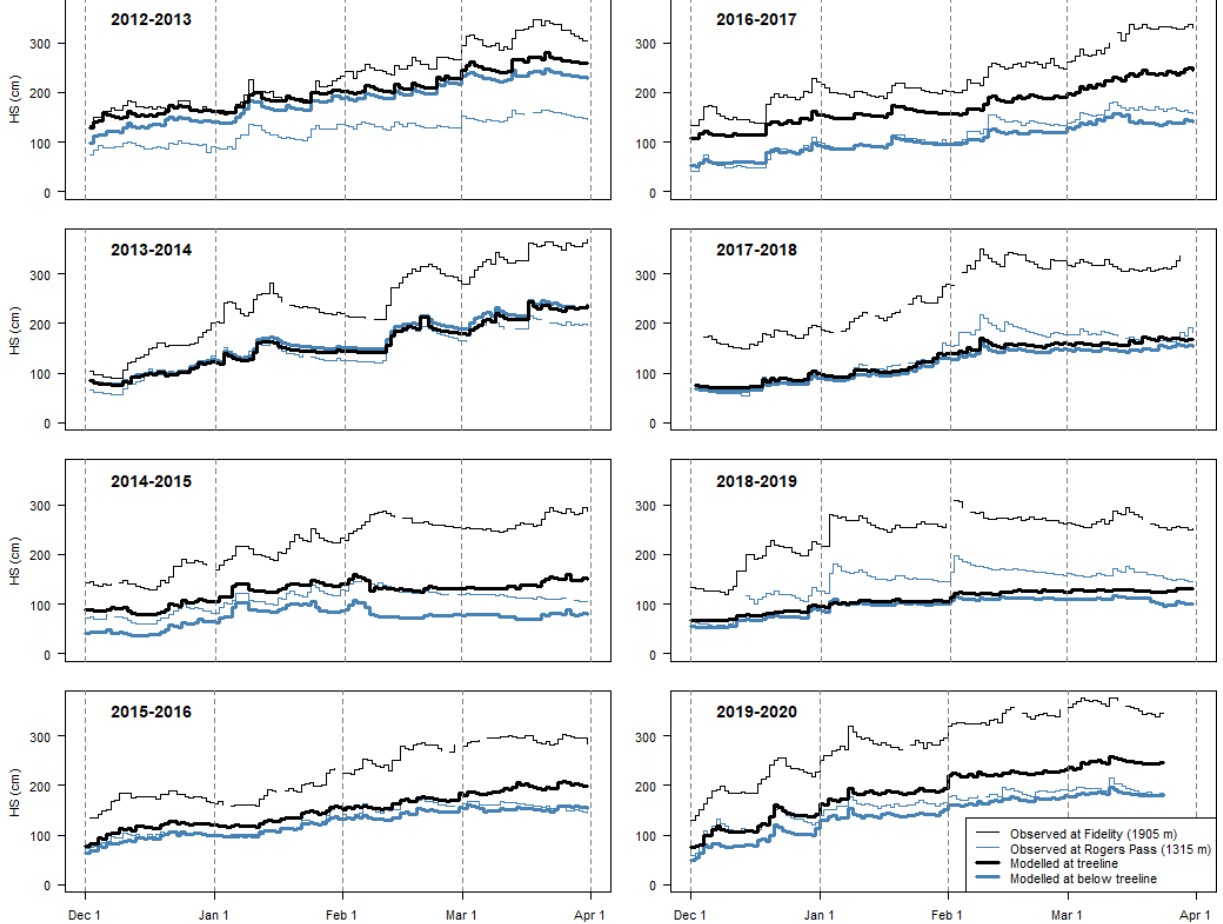

**Figure 4.** Modelled snow height (HS) at treeline and below treeline grid points in Glacier National Park compared to field observations from study plots at similar elevations.

at treeline and below treeline, respectively. The overall accuracy of predicting days with more than a trace of new snow was 86% (i.e. HN24 > 0.1 cm). Days with more than a trace amount of snowfall were predicted with a hit rate of 91% and a false alarm rate of 10%, while days with no snowfall or a trace amount were predicted with a hit rate of 77% and a false alarm rate of 21%. Based on the quantity and timing of modelled precipitation, the model-generated weather and snowpack variables are expected to capture the majority of weak layer and slab formation events over the study period, but also likely contain inaccuracies about the detailed properties of the slabs and weak layers.

## 3.2 Storm slab avalanche problems

When only considering weather and snowpack variables, the most significant variable to explain the presence of storm slab avalanche problems was the height of new snow over the past 72 h (Fig. 5). Storm slab problems were present 62% of the time





**Table 5.** Verification of modelled snow height and snowfall with field observations.

| | Spearman correlation coefficient | | Percent difference (model relative to observed) | |
| --- | --- | --- | --- | --- |
| | Treeline | Below treeline | Treeline | Below treeline |
| Snow height | 0.89 | 0.89 | -36% | -5% |
| 72 h snowfall | 0.82 | 0.86 | -14% | +40% |
| 24 h snowfall | 0.78 | 0.82 | -5% | +28% |

when the height was more than 7 cm and 77% of the time when the height was more than 16 cm. The next most significant variable was the maximum hourly wind speed over the past 24 h, with storm slabs becoming more common when small amounts of new snow coincided with higher wind speeds. Other variables appearing lower in the decision tree included the grain type beneath the last 72 h of new snow (storm slab problems were more common when this was rounded grains), the

average slab density of snow from the past 72 h (storm slab problems were more common when the density was greater than 113 kg m$^{-3}$), and the maximum intensity of incoming shortwave radiation in the past 72 h (storm slab problems more common when shortwave radiation was weaker).

When adding contextual variables, the first and most significant variable was whether a storm slab problem was present the previous day (Fig. 6), with the problem persisting 81% of the time. The subsequent splits describe when storm slab problems

were newly added to a hazard assessment (i.e. left branch of the tree) or when storm slab problems were removed (i.e. right branch of the tree). The most significant rule for adding a new storm slab problem was height of new snow over the past 24 h, with a problem being added 35% of the time when there was more than 5 cm of new snow. The most significant rule for removing a storm slab problem was the amount of incoming longwave radiation over the past 24 h, with the problem more frequently removed after a period of clear skies (i.e. lower values of incoming longwave radiation). When adding a new storm

slab problem, the second most important variable after a threshold amount of new snow was whether there was already a wind slab problem in the hazard assessment, with very few cases of both storm and wind slab problems present on the same day. Wind slab problem variables on the right branch of the decision tree show a similar theme of wind slab problems replacing storm slab problems after a period of clear skies.

When using the decision trees for classification, the addition of contextual variables increased the percentage of correct

classifications from 78% to 88% (Table 6), increased the hit rate from 76% to 84%, and decreased the false alarm rate from 35% to 13%.

### 3.3 Wind slab avalanche problems

When only considering weather and snowpack variables, the most significant variable for wind slab avalanche problems was the maximum air temperature over the past 24 h (Fig. C1). Wind slab problems were more common when the maximum air

temperature was less than -9 ° C. Snowpack variables in the decision tree included the grain type beneath the last 72 h of new snow (wind slab problems were more common when the grain type was precipitation particles, rounded grains, or faceted

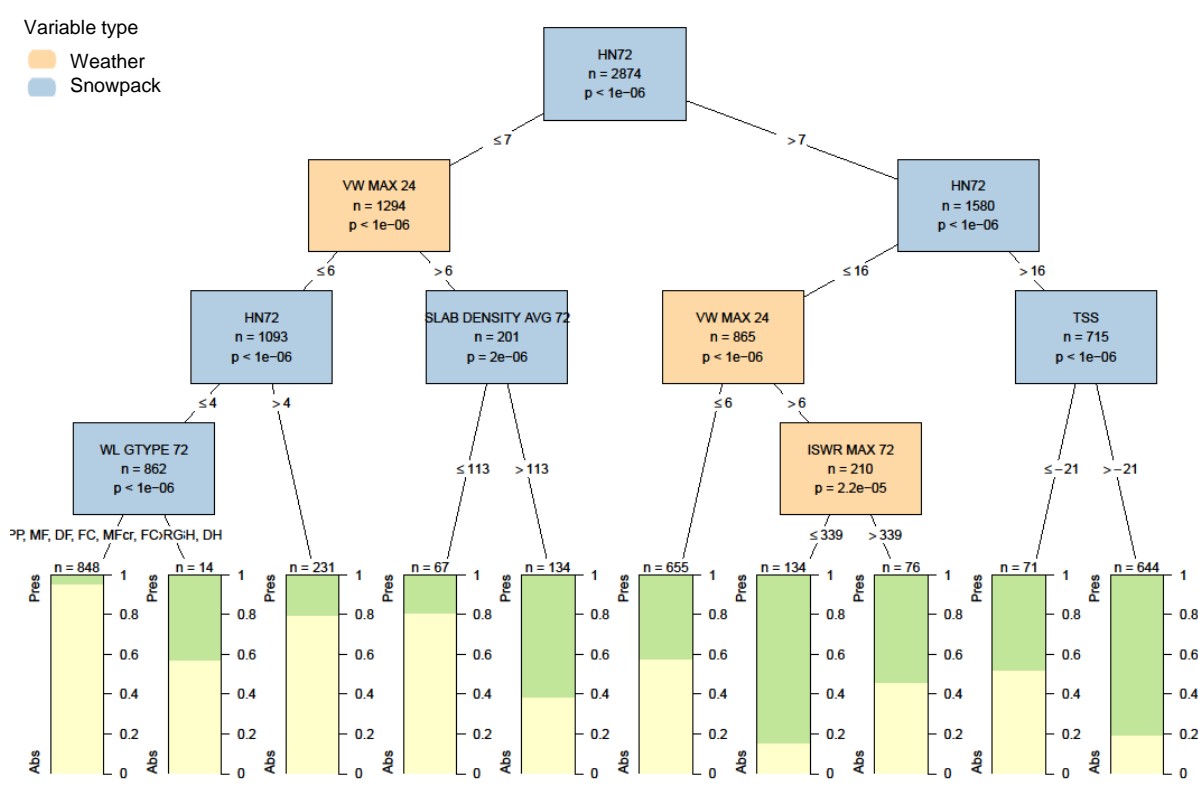

**Figure 5.** Decision tree for the presence or absence of storm slab avalanche problems based on weather and snowpack variables. Variables appearing in the tree include the height of snow in the past 72 h in cm (HN72), the maximum hourly wind speed over the past 24 h in m s$^{-1}$ (VW MAX 24), the grain type beneath snow deposited in the past 72 h (WL GTYPE 72), the average density of all snow deposited in the past 24 h in kg m$^{-3}$ (SLAB DENSITY AVG 72), the maximum intensity of incoming shortwave radiation in the past 72 h in MJ m$^{-2}$ (ISWR MAX 72), and the snow surface temperature in ° C (TSS). Terminal nodes show the proportion of cases with storm slab problems present (green) and absent (yellow).


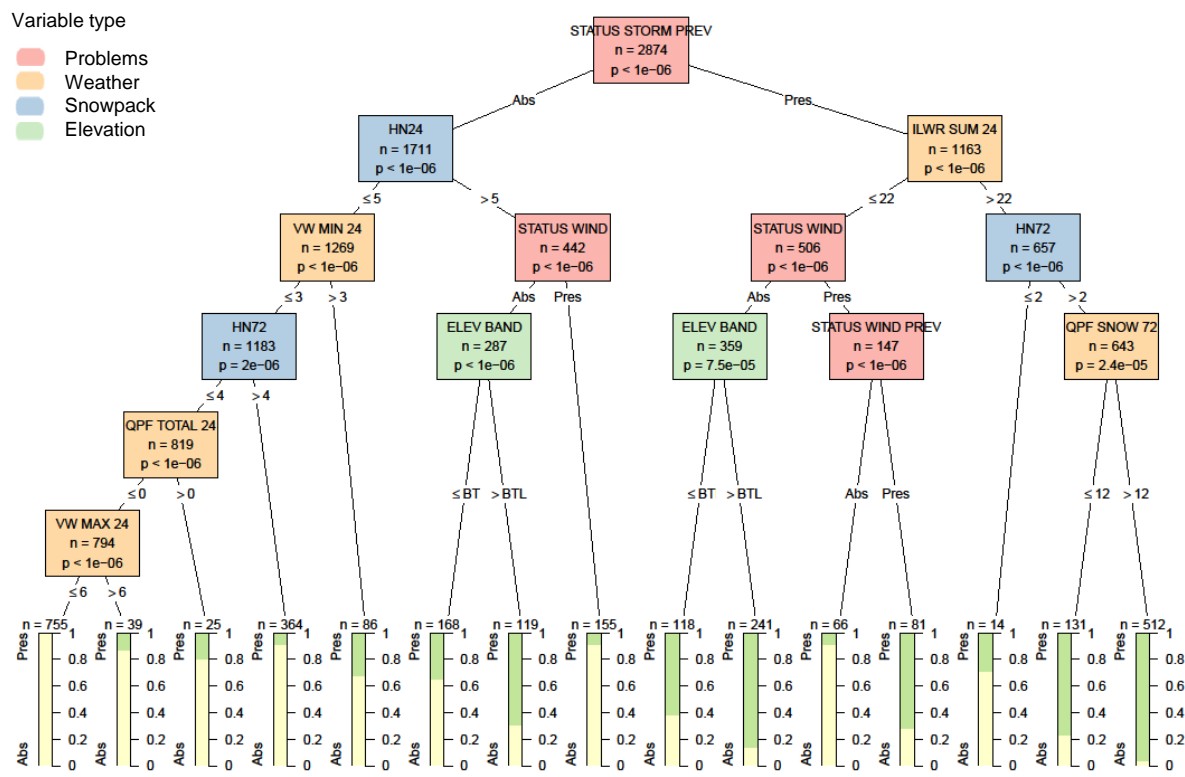

**Figure 6.** Decision tree for the presence or absence of storm slab avalanche problems based on weather, snowpack, and contextual variables about the hazard assessment. Variables appearing in the tree include the presence or absence of a storm slab problem the previous day (STATUS STORM PREV), the height of snow in the past 24 h in cm (HN24), minimum hourly wind speed over the past 24 h in m s$^{-1}$ (VW MIN 24), the total amount of precipitation in the past 24 h in mm (QPF TOTAL 24), the maximum hourly wind speed over the past 24 h in m s$^{-1}$ (VW MAX 24), the presence or absence of a wind slab problem the same or previous day (STATUS WIND and STATUS WIND PREV), elevation band (ELEV BAND), the total incoming longwave radiation accumulated over the past 24 h in MJ m$^{-2}$ (ILWR SUM 24), and the total amount of solid precipitation over the past 72 h in mm (QPF SNOW 72). Terminal nodes show the proportion of cases with storm slab problems present (green) and absent (yellow).





**Table 6.** Contingency table statistics when classifying the presence of surface avalanche problem types with decision tree models.

| Problem type | Percent of cases with problem present | Accuracy | | Hit rate | | False alarm rate | |
|---|---|---|---|---|---|---|---|
| | | Weather and snowpack variables only | Additional contextual variables included | Weather and snowpack variables only | Additional contextual variables included | Weather and snowpack variables only | Additional contextual variables included |
| Storm slab | 41% | 77 | 88 | 76 | 84 | 35 | 13 |
| Wind slab | 31% | 74 | 92 | 59 | 85 | 50 | 12 |
| Dry loose | 15% | 85 | 89 | - | 64 | - | 35 |
| Wet loose | 7% | 94 | 96 | 90 | 69 | 83 | 35 |
| Cornice | 3% | 97 | 98 | - | 74 | - | 29 |

crystals), the size of any surface hoar on the surface (wind slab problems were less common when there was large surface hoar on the surface), the height of new snow since the last snow free day, the grain type that formed before the last snow free day, and the total height of the snowpack.

When adding contextual variables, the first and most significant split was whether a wind slab problem was present the

5 previous day (Fig. C1), with the problem persisting 82% of the time. The variables that contributed to adding a new wind slab problem included elevation band (wind slab problems were uncommon at the below treeline band), the presence or absence of a storm slab problem the same or previous day, the particular season, and the time of year (wind slab problems were more common before 7 January). The variables that contributed to removing a wind slab problem were the presence or absence of a storm slab problem the same or previous day and the size of any surface hoar on the surface (wind slabs were removed more

frequently when surface hoar had grown larger than 1.9 mm).

When using the decision trees for classification, the addition of contextual variables increased the percentage of correct classifications from 74% to 92% (Table 6), increased the hit rate from 59% to 85%, and decreased the false alarm rate from 50% to 12%.

### 3.4 Dry loose avalanche problems

The most significant variable impacting dry loose avalanche problems in the decision trees with only weather and snowpack variables was the skier penetration depth (Fig. C2). Dry loose problems were present 19% of the time when the skier penetration depth was greater than 18 cm. The only other significant variable was the average density of snow deposited since the last snow free day, with dry loose problems more common when the average slab density was less than 139 kg m$^{-3}$. When contextual variables were included in the analysis, dry loose problems persisted 64% of time when they were present the previous day.





The variables that influenced adding a new dry loose problem included skier penetration depth, the maximum air temperature over the past 72 h, and the size any of surface hoar on the surface. The decision tree algorithm did not identify any variables significantly associated with the removal of a dry loose problem.

Using the decision tree with only weather and snowpack variables for classification would never predict a dry loose avalanche problem, because all terminal nodes had a small proportion of cases with the problem present. Accordingly, since dry loose problems were absent in 85% of the cases the tree had an accuracy of 85% (Table 6). The decision tree with contextual variables would only classify a dry loose problem if the problem was present the previous day, which increased the accuracy to 89%. However, since dry loose problems were rare events, the hit rate and false alarm rate were relatively poor.

### 3.5 Wet loose avalanche problems

When only considering weather and snowpack variables, wet loose avalanche problems were primarily influenced by air temperature, with the problem becoming present more frequently at warmer temperatures (Fig. C3). In this analysis, almost all splits were threshold amounts for the maximum air temperature over the past 24 or 72 h. One snowpack variable appearing in the decision tree was whether there was a melt-freeze crust beneath the snow deposited in the past 72 h. In the decision tree with the contextual variables, wet loose problems persisted 69% of the time and, like dry loose problems, had no significant variables impacting the removal of the problem. The variables for adding a new wet loose problem included the maximum air temperature over the past 72 h and whether it was before or after 16 March (wet loose problems were more likely added in the late winter).

Similar to dry loose problems, the decision trees for wet loose problems also had relatively high classification accuracy because the problem was absent the majority of time (Table 6). The tree using temperature thresholds had a better hit rate than the tree using the status of wet loose problems the previous day (90% versus 69%), but also resulted in a very high false alarm rate (85%).

### 3.6 Cornice avalanche problems

The analysis of only weather and snowpack variables found significant variables for cornice avalanche problems were the total height of the snowpack (Fig. C4), maximum of shortwave radiation over the past 24 h, maximum hourly wind speed over the past 24 h, the height of new snow since the last snow free day, and the snow surface temperature. When including contextual variables, persistence played a dominant role in the presence of cornice problems, with cornice problems persisting 74% of the time and no significant variables for removing a cornice problem. The most significant variable influencing when to add a new cornice problem was elevation band (cornices were primarily present in alpine elevation band) and then a secondary split based on the maximum shortwave radiation over the past 24 h (cornice problems were added with large amounts of incoming shortwave radiation).

The decision tree with weather and snowpack variables never resulted in situations when cornice problems would be classified, but the decision tree with contextual variables would predict a cornice problem if it was present the previous day. Since cornice problems were very rare, these decision trees had high accuracies (97% and 98%).




### 3.7 Persistent avalanche problem types

The 54 weak layers attributed to persistent avalanche problem types were found to have significantly larger grain sizes than another 41 potential weak layers that were also described in the hazard assessments but did not result in persistent problems. A decision tree model comparing these two sets of layers found layers with grain sizes larger than 1.1 mm were more likely to

become attributed to persistent problems (not shown).

The most significant variable influencing a known weak layer's association with persistent avalanche problems was the density of the slab above the weak layer. When analyzing all the cases for the 54 weak layers from the time they were buried until one week after they were no longer associated with a persistent problem found these layers were associated with persistent problems 66% of the time when the maximum slab density was greater than 214 kg m$^{-3}$, compared to only 15% of the time

when the slab density was below that threshold (Fig. 7). Weak layer age was the second most influential variable, appearing several places in the decision tree. The situations when these weak layers were most often associated with persistent problems were when the slab density was relatively high and when the weak layer was between 6 and 26 d old. The most common situations when these weak layers had not yet developed into persistent problems (i.e. *surface* status) were when the slab density was low and when weak layers were young (with threshold ages of 6, 8, and 18 d appearing in the tree). Two additional

situations delayed the onset of persistent problems; periods of low incoming solar radiation and when there was a low-density weak layer with more than 21 cm of snow over the past 72 h. These situations both describe conditions that would be expected during storms, suggesting the weak layer would more likely be associated with a surface problem type. The most common situations when these weak layers were no longer associated with persistent problems (i.e. *dormant* status) were when they were older (with threshold ages of 18 and 26 d appearing in the tree) and after periods of sustained wind (minimum hourly

wind speed greater than 0.4 m s$^{-1}$ over the past 72 h).

The dominant conditions related to forecasters decisions to add or remove persistent problems become clearer when narrowing the focus to the actual start and end of persistent problems. Due to the smaller size of the data sample, these analyses revealed fewer significant variables at the p = 0.0001 level, so the maximum p-value for variables shown in the decision trees was increased to 0.05. Maximum slab density was again the most significant difference between the three days before layers became

associated with persistent problems and their first three days of being associated with persistent problems (Fig. 8). Exceeding a maximum slab density of 193 kg m$^{-3}$ increased the frequency of the layer being a problem from 32% to 57%. Snow surface temperatures colder than -18 ° C further increased the frequency of weak layers becoming associated with persistent problems.

Comparing conditions before and after weak layers became dormant revealed no significant difference for time periods of 3 or 7 d before and after the transition. Expanding the window to compare conditions two weeks before and after the transition

revealed several significant differences (Fig. 9). First, weak layers were more commonly associated with a persistent problem when the maximum slab density was less than 292 kg m$^{-3}$. Additional factors that contributed to weak layers persisting were an age less than 16 d, and air temperatures above -8 ° C. The slab density above weak layers was the most significant variable for both the onset and the end of persistent problems, with weak layers most often associated with persistent problems when the maximum slab density was between 193 and 292 kg m$^{-3}$.

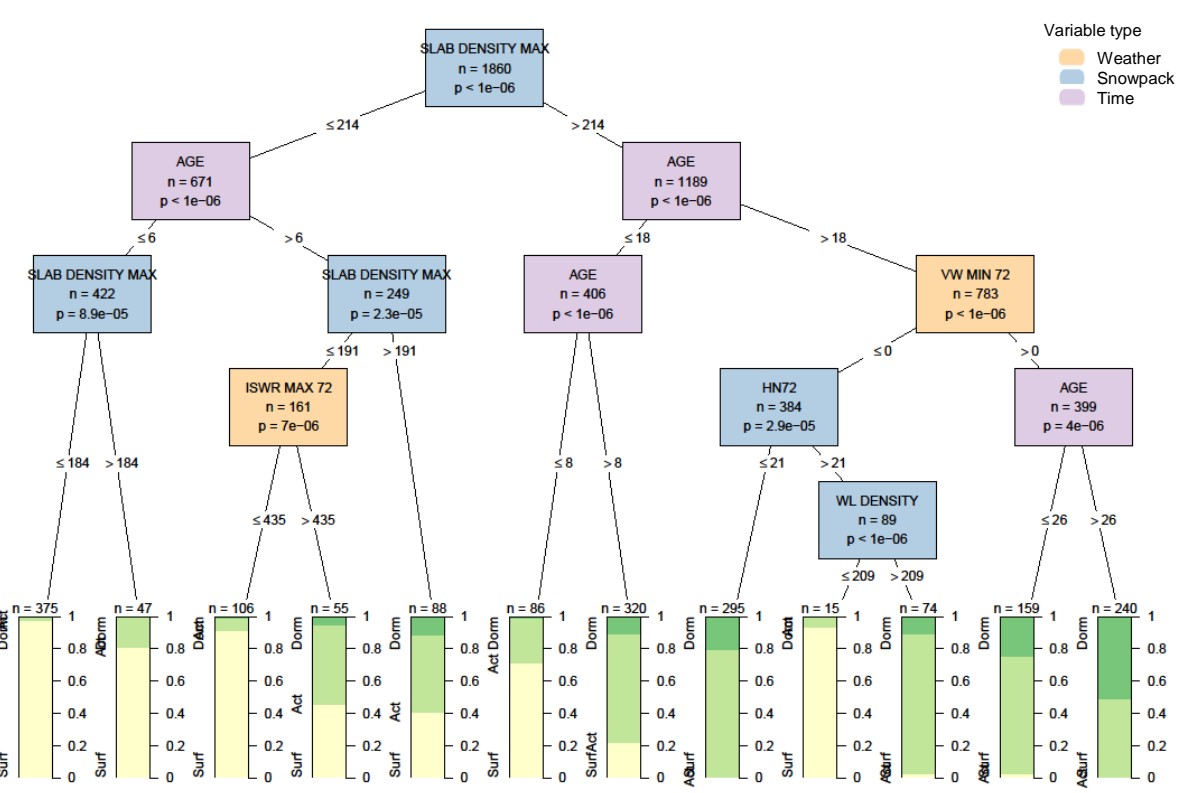

**Figure 7.** Decision tree for weak layers developing into a persistent avalanche problem types as their status transitions from *surface* to *active* to *dormant*. Variables appearing in the tree include the maximum slab density in kg m$^{-3}$ (SLAB DENSITY MAX), the age of the weak layer in days (AGE), the maximum intensity of incoming shortwave radiation in the past 72 h in MJ m$^{-2}$ (ISWR MAX 72), the minimum hourly wind speed over the past 72 h in m s$^{-1}$ (VW MIN 72), the height of snow in the past 72 h in cm (HN72), and the weak layer density in kg m$^{-3}$ (WL DENSITY). Terminal nodes show the proportion of cases with weak layers having a *surface* status (yellow), *active* status (light green), and *dormant* status (dark green).





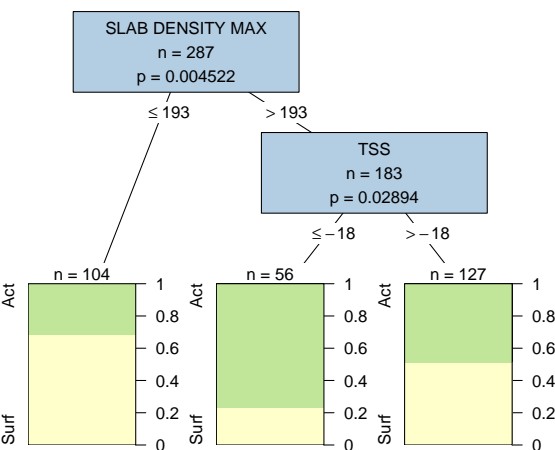

**Figure 8.** Decision tree comparing conditions before and after weak layers first become associated with persistent problems (considering three days before they become problems and the first three days layers of being associated with a problems). Variables appearing in the tree include the maximum slab density in kg m$^{-3}$ (SLAB DENSITY MAX) and the snow surface temperature in ° C (TSS). Terminal nodes show the proportion of cases with weak layers having *surface* (yellow) and *active* (green) statuses.

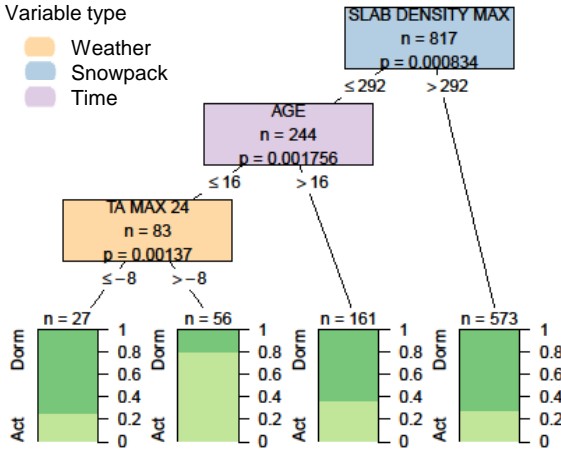

**Figure 9.** Decision tree comparing conditions before and after weak layers become dormant (considering the first 14 days when layers are no longer associated with persistent problems with the last 14 days when they are associated with persistent problems). Variables appearing in the tree include the maximum slab density in kg m$^{-3}$ (SLAB DENSITY MAX), the age of the weak layer in days (AGE), and the maximum air temperature over the past 24 h ° C (TA MAX 24). Terminal nodes show the proportion of cases with weak layers having *active* (light green) and *dormant* (dark green) statuses.



## 4   Discussion

### 4.1   Lessons about application of avalanche problems

The decision tree analyses with contextual information and those with only weather and snowpack information offer two different perspectives of how avalanche problems are applied. The weather and snowpack variables offer a snow science perspective into what conditions favoured the formation of different avalanche problem types, while the contextual variables show how forecasters integrated weather and snowpack information into their hazard assessment process. The decision trees with contextual variables always started by asking whether the problem was present the previous day, illustrating an iterative Bayesian process where each assessment starts with the prior assessment and then considers new information (LaChapelle, 1980).

The addition of contextual variables resulted in decision tree models that better represented the original data, largely due to the prominent role of persistence. The improvement was indicated by a higher accuracy, higher hit rates, and lower false alarm rates when using the trees for classification (Table 6). For example, when adding contextual variables, the percentage of correct classifications by the storm slab decision tree increased from 78% to 88%, the hit rate increased from 76% to 84%, and false alarm rate dropped from 35% to 13%. The decision tree for wind slab problems had similar improvements when adding contextual variables. The improvements were less clear for dry loose, wet loose, and cornice avalanche problems because these problems were less prevalent in the dataset. Even with statistically significant variables in their decision trees, the terminal nodes still suggest these problems were absent most of the time, unless it was known that the problem was present the previous day. This also reinforces the important role of persistence in avalanche forecasting.

Another dominant theme was adding new avalanche problems often had more significant and intuitive explanations than removing old avalanche problems, suggesting problems were added more consistently by forecasters than they were removed. This is evident by the greater number of significant variables in the left branch of trees with contextual variables (i.e. situations when a problem would be added) than in the left branch of these trees (i.e. situations when a problem would be removed). For example, there were no significant weather or snowpack variables to explain the removal of dry loose, wet loose, or cornice avalanche problems. Similarly, when analyzing the onset of persistent avalanche problems there were several significant differences in conditions three days before and after the onset of the problem (Fig. 8), but there were no significant differences three or even seven before and after the end of the problem. These patterns suggest forecasters likely have greater confidence, precision, and consistency when adding new problems. Removing problems could be more inconsistent due to uncertainty about when the likelihood or consequence of avalanches has reduced to the point where problems no longer need to be listed (Lazar et al., 2012). However, it is important to remember that our analysis did not include the full range of observations that avalanche forecasters consult when removing an avalanche problem (e.g. avalanche observation data).

Significant interactions between storm slab and wind slab avalanche problems were apparent in the decision trees (Fig. 5 and Fig. C1b), as at least one of these two problems were listed in the hazard assessment on 83% of the days over the study period. Some operational policies suggest using only one of these two problems at a time to effectively communicate the most relevant risk management approach (Klassen, 2014). The decision trees with contextual variables show this practice



was applied with wind slabs typically being removed when storm slab problems were added and then reintroduced when storm slab problems were removed. The most intuitive rule based on weather or snowpack condition from these trees was adding storm slab problems when a threshold amount of snowfall occurred. However, the variables for removing storm slab problems and adding or removing wind slab problems had less intuitive interpretations (e.g. incoming longwave radiation

and air temperature). One possible explanation is that storm slab problems are more directly influenced by changing weather conditions while wind slab problems exist most of the winter but are removed when they are masked by more important storm slab problems. Another possible explanation of the observed pattern is that the model-generated data did not capture all the processes that form wind slab problems.

The significance of variables related to the time and location of the hazard assessment also highlight forecasting practices

that may override physical weather and snowpack conditions. For example, wind slab problems were more likely before 7 January, wet loose problems were more likely after 16 March, and wind slab and cornice problems were more likely at higher elevations. While these patterns have intuitive explanations, contextual variables were more significant than physical explanations that could have been captured with weather and snowpack variables. For example, wind speed and temperature variables should capture stronger wind at higher elevations and warmer conditions later in the season with more nuance than

rules based on elevation and day of season. This result seems to indicate that forecasters may apply heuristics when uncertain about the physical conditions, such as adding wind slab problems in the alpine because they are common in the alpine. Some problems may also be prioritized differently in certain situations, for example, wind slab problems may become lower priority as the season progresses because other higher consequence avalanche problems develop and take priority.

## 4.2   Lessons about forecasting with weather and snowpack models

Variables generated by weather and snowpack models provided intuitive explanations for most avalanche problem types, despite discrepancies between modelled and observed conditions. For example, although the models systematically underestimated precipitation, the most significant variables influencing storm slab problems were threshold amounts of new snow (Fig. 4 and 5). The threshold amount of 7 cm over 72 h in Fig. 4 was less than expected from practical experience, as Schirmer et al. (2009) found a 12 cm threshold for 72 h snowfall as a strong predictor of avalanche danger and McClung and Schaerer (2006) suggest

30 cm of new snow as a rule of thumb for natural avalanche activity. Accordingly, the types of variables appearing in the decision trees are more important than the specific thresholds, which should be treated with caution when compared to values expected from field observations.

The model-generated variables were more intuitive for some avalanche problem types than others. The most significant variables were intuitive for storm slab problems (height of new snow in past 72 h), dry loose problems (skier penetration

depth), wet loose problems (maximum air temperature), and persistent slab problems (slab density). The most significant variables for wind slab and cornice problems were air temperature and snow height , respectively, which seem less connected to the formation of these problems. Variables directly associated with snow transport by wind would have been more intuitive, suggesting the model configuration likely did not adequately capture snow transport processes. Improved prediction of wind





slab and cornice problems could be achieved with model configurations that better capture snow transport (e.g. Vionnet et al., 2018).

Models also offer opportunities to capture conditions that are difficult to measure in the field. For example, slab density above a weak layer was found to have a significant influence on persistent avalanche problem types. Slab densification is
difficult to continually quantify from field observations but is likely a process that forecasters hypothesize about when assessing persistent problem types. Models could help forecasters explicitly test hypotheses about slab density trends and similar physical properties. Although this analysis used weak layers identified by forecasters, weak layer detection could also be automated by models (Monti et al., 2014). For example, the 54 weak layers in this analysis were found to have significantly larger grain sizes than other potential weak layers over the same period, suggesting layer properties could be analyzed to detect potential weak
layers. However, our approach was more appropriate for understanding how forecasters applied persistent problem types.

### 4.3   Applications and limitations for decision support tools

Data-driven decision support tools like the decision trees presented in this study have the potential to improve the accuracy and consistency of avalanche hazard assessments (Clark, 2019; Lazar et al., 2016; Statham et al., 2018b; Techel et al., 2018). Providing guidance on avalanche problems could provide more tangible information to forecasters about necessary risk
mitigation measures than previous decision aids that focused on danger level or avalanche occurrence. Expanding the decision tools to provide insight into additional avalanche problem attributes such as the location, size, and likelihood of avalanches could offer additional value.

While this study focused on model-generated weather and snowpack variables due to their consistency and temporal coverage, decision support tools could also include a broader range of inputs that represent the data available to forecasters more
comprehensively. First, including field observations of weather and snowpack conditions would be a meaningful complement to overcome issues with model errors. Second, including avalanche activity data would likely improve aspects of the decision trees such as providing better explanations of when avalanche problems are removed. A challenge would be formatting field observations in a way that allows them to be integrated into a statistical model in a consistent way. Future decision aids should consider leveraging the respective strengths of field observations and model-generated variables.

Using operational datasets to build predictive models will continue to face issues with messy and inconsistent patterns that could perpetuate errors in human assessments (Guikema, 2020). The decision trees presented in this study only showed the most significant variables by restricting the p-value to 0.0001 to keep the diagrams tidy and easy to interpret. However, adjusting the maximum p-value to 0.05 revealed decision trees with many additional significant variables that become more challenging to interpret. While such models may offer considerable predictive power, the underlying rules should be grounded in physical
conditions and operational priorities to provide optimal and unbiased decision support.


## 5  Conclusions

We employed conditional inference trees to examine patterns in how avalanche forecasters combine different types of weather and snowpack conditions into their assessment of avalanche problems. We used simulated weather and snowpack observations to create a consistent dataset and explore the potential of physical models to complement traditional field observations.

Some problems had clear and intuitive rules about specific conditions that explained their presence in hazard assessments, such new snow thresholds for storm slab avalanche problems and slab density ranges for persistent slab avalanche problems. Other problems appeared to be more influenced by contextual information such as time of season, location, and presence or absence of other avalanche problems. Wind slab avalanche problems, for example, were better explained by the absence of storm slab problems than by weather and snowpack conditions. In addition, we also found that the patterns for adding new

avalanche problems to hazard assessments were much clearer compared to when they were removed, highlighting potential inconsistencies in forecasting practices.

Our analysis highlights that avalanche problem assessments included in public avalanche bulletins are not only based on patterns in weather conditions and snowpack structure, but might also be influenced by human risk management needs and risk communication priorities. The present analysis provides a systematic approach for examining how the different perspectives

affect existing forecasting practices. The decision trees emerging from our analyses provide a concise and tangible summary of existing practices that can be used to facilitate expert discussions about current practices and how to increase consistency within forecasting teams and across agencies.

While it might be tempting to use the decision trees presented in this study as decision support tools for operational avalanche forecasting, it is important to remember that the identified decision rules are affected by errors and biases in both the past

assessments and the simulated data. Hence a purely data-driven approach to the design of decision aids will perpetuate past errors and biases. To avoid this issue, we propose a mixed method approach that combines insights from data driven analysis methods with qualitative studies on avalanche forecasting practices. The combination of the two perspectives will lead to a better understanding of existing practices and lay the foundation for meaningful decision aids that are grounded in snow science principles but also take advantage of the human avalanche forecasting expertise.

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

**Notes**

. The code and data are publicly available at https://osf.io/ypzhe.

10   . All authors conceptualized the research. MT did the initial analysis and writing for a pilot project, which was expanded upon for this paper by SH. PH provided supervision and acquired the funding.

. The authors declare that they have no conflict of interest.

. Thanks to Avalanche Canada and the Avalanche Canada Foundation.





## Appendix A: Contingency table statistics

A contingency table shows the combined frequencies of observed and modelled values. For a dichotomous variable (i.e., yes or no) the contingency table has four possible combinations of observed and modelled values (Table A1). Numerous statistics describe the marginal distributions in contingency tables, in this analysis we report accuracy, hit rate, and false alarm rate.

5  Accuracy is the fraction of modelled values that were correct:

$$Accuracy = (hits + correct\ negatives)/total \tag{A1}$$

Hit rate is the fraction of observed yes events that were correctly modelled:

$$Hit\ rate = hits/(hits + misses) \tag{A2}$$

False alarm rate is the fraction of modelled yes events that did not occur:

10  $$False\ alarm\ rate = false\ alarms/(hits + false\ alarms) \tag{A3}$$

**Table A1.** Contingency table.

|  |  | Observed | |
|---|---|---|---|
|  |  | Yes | No |
| Modelled | Yes | hits | false alarms |
|  | No | misses | correct negatives |



## Appendix B:  Correlations between weather and snowpack variables



**Figure B1.** Correlation matrix for weather and snowpack variables used in decision trees for surface avalanche problems. Variables in bold were removed due to high correlation with other variables. Variable names are described in Tables 2 and 3.

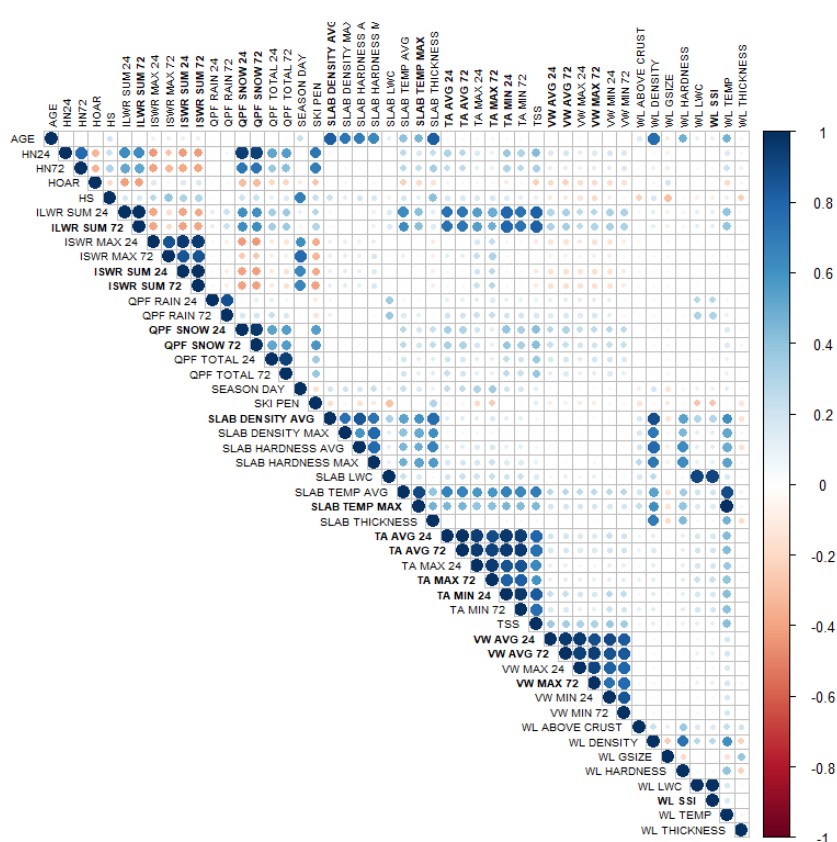

**Figure B2.** Correlation matrix for weather and snowpack variables used in decision trees for persistent avalanche problems. Variables in bold were removed due to high correlation with other variables. Variable names are described in Tables 2 and 3.





## Appendix C: Additional decision trees for surface avalanche problem types

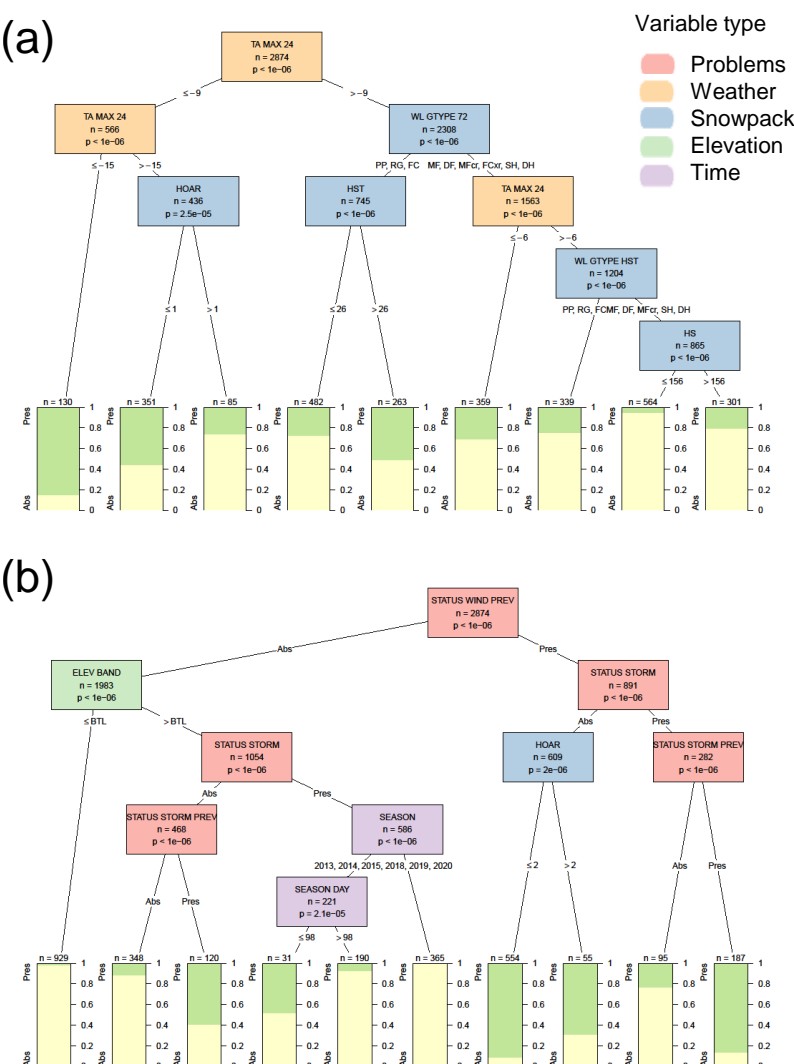

**Figure C1.** Conditional inference tree for the presence or absence of wind slab avalanche problems (a) based on weather and snowpack variables and (b) including additional contextual variables about the hazard assessment. Variables appearing in the tree include maximum air temperature over the past 24 h in ° C (TA MAX 24), the size of surface hoar on the surface in mm (HOAR), the grain type beneath snow deposited in the past 72 h (WL GTYPE 72), the height of new snow since the last snow free day (SLAB THICKNESS HST), the grain type beneath snow deposited since the last snow free day (WL GTYPE HST), the total snow height in cm (HS), the presence or absence of a wind slab problem the previous day (STATUS WIND PREV), elevation band (ELEV BAND), the presence or absence of a storm slab problem the same or previous day (STATUS STORM and STATUS STORM PREV), the specific season (SEASON), and the number of days since 1 October (SEASON DAY). Terminal nodes show the proportion of cases with wind slab problems present (green) and absent (yellow).

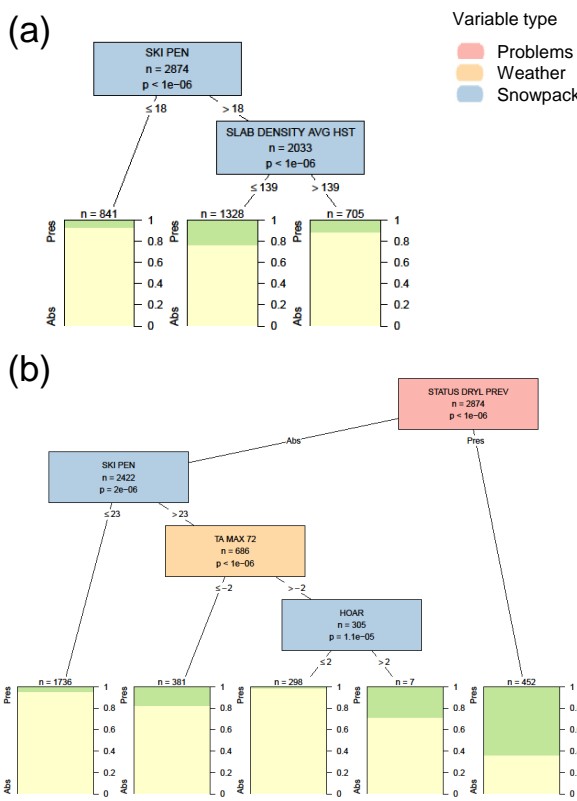

**Figure C2.** Conditional inference tree for the presence or absence of dry loose avalanche problems (a) based on weather and snowpack variables and (b) including additional contextual variables about the hazard assessment. Variables appearing in the trees include the skier penetration depth in cm (SKI PEN), the average density of new snow since the last snow free day in kg m⁻³ (SLAB DENSITY AVG HST), the presence or absence of a dry loose problem the previous day (STATUS DRYL PREV), maximum air temperature over the past 72 h in °C (TA MAX 72), and the size of surface hoar on the surface in mm (HOAR). Terminal nodes show the proportion of cases with loose dry problems present (green) and absent (yellow).



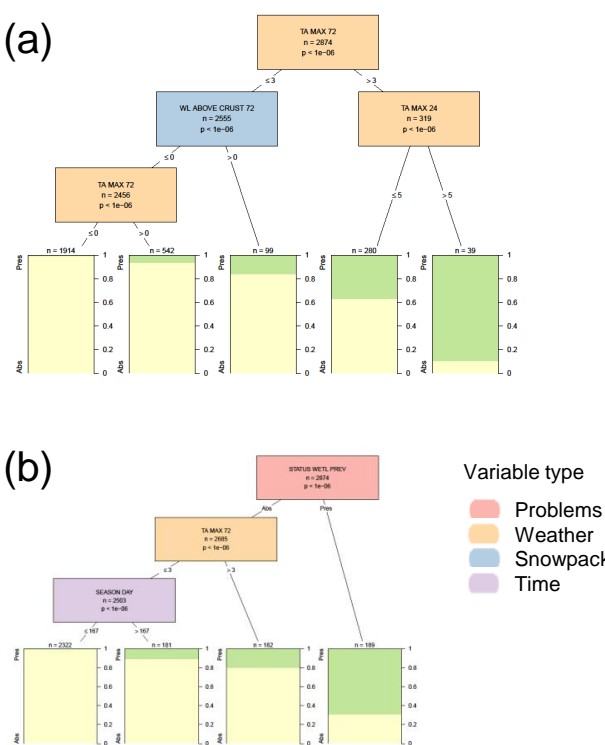

**Figure C3.** Conditional inference tree for the presence or absence of wet loose avalanche problems (a) based on weather and snowpack variables and (b) including additional contextual variables about the hazard assessment. Variables appearing in the trees include the maximum air temperature over the past 24 and 72 h (TA MAX 24 and TA MAX 72), whether there was a melt-freeze crust beneath the snow deposited in the past 72 h (WL ABOVE CRUST 72), the presence or absence of a wet loose problem the previous day (STATUS WETL PREV), and the number of days since 1 October (SEASON DAY). Terminal nodes show the proportion of cases with wet loose problems present (green) and absent (yellow).


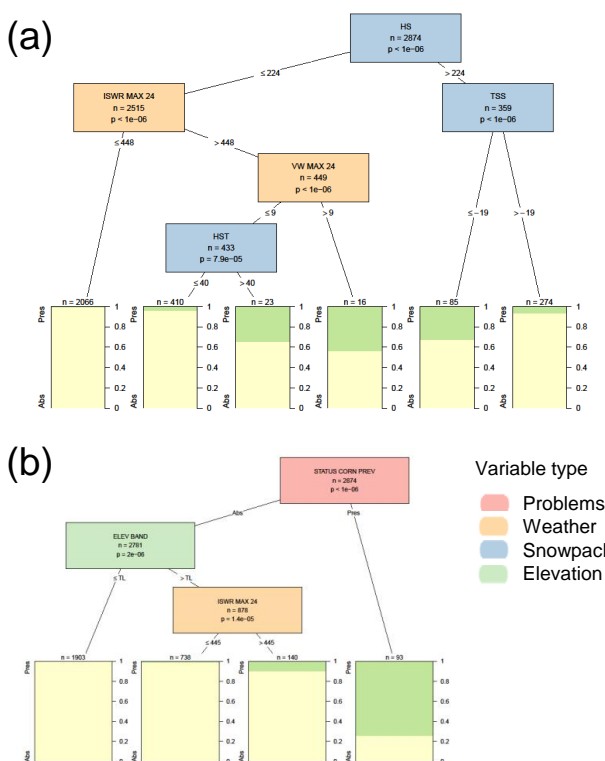

**Figure C4.** Conditional inference tree for the presence or absence of cornice avalanche problems (a) based on weather and snowpack variables and (b) including additional contextual variables about the hazard assessment. Variables appearing in the trees include the total snow height in cm (HS), the maximum incoming shortwave radiation in the past 24 h in W m$^{-2}$ (ISWR MAX 24), maximum hourly wind speed in the past 24 h in m s$^{-1}$ (VW MAX 24), the height of new snow since the last snow free day (HST), snow surface temperature in °C (TSS), the presence or absence of a cornice problem the previous day (STATUS CORN PREV), and the elevation band (ELEV BAND). Terminal nodes show the proportion of cases with cornice problems present (green) and absent (yellow).