# Peer review of "Examining the operational use of avalanche problems with decision trees and model-generated weather and snowpack variables"

_Natural Hazards and Earth System Sciences, 2020_

## Referee Comment (RC1) · Rune Engeset (Referee) · 18 Sep 2020

**Review of https://doi.org/10.5194/nhess-2020-274**

**General comments**

I really enjoyed reading this manuscript. It provided scientific value, by analysing forecast and snow/weather model data in a thorough and novel manner with the view of providing applied value to avalanche risk forecasting and management. It applies decision tree analysis to avalanche problems, based on simulated snow and weather properties. The data are abundant, documented, structured and relevant, the analysis and methods are well described and pertinent to the problem, and the focus on the avalanche problems adds significant new knowledge to the research and operational communities. The presentation of the results is adequate and the discussion clear and to the point. In general, the figures and tables are easy to read. It could add value to discuss the results in a more generalised context, please consider if any of these points are relevant based on your data and analysis: How may the results be transferable to other warning services and other climatic settings? How would the results be if the four (five) EAWS problems were used instead of the more detailed North American types? What about using the forecaster name/ID and the day number in his/her forecasting period as variables, was this tested?

I recommend publication of the paper, after minor revision addressing the detailed comments below.

**Detailed comments (p refers to page number and # to line number)**

p2#35 In the introduction, it could be mentioned that the utility of snow model data in operational forecasting is to a certain extent hampered by the quality and representativeness of input weather data. If a weather model or weather station fails to catch a precipitation event, this error will propagate to the snow model simulation and it may linger on for a long period.

p3#22 Explain why this period (winter months with dry snow avalanche as primary concern) is chosen. Consider adding more details about the study area, e.g. latitude, tree line altitude and elevation interval or hypsometry.

p3#16 Consider adding "and time period" to the sub-title

p4Fig1 The sentence "Grid points used before the 2017 model update are shown with circles and grid points used after the update are shown with squares." in the caption makes limited sense to the reader at this stage in the text. Could you explain better?

p11#25 Could you add a summary of selected variables? It may improve the reader' ability to catch up in the text.

p12#4-5 It is not only precipitation controlling weak layer and slab formation events, thus this sentence could be moderated or elaborated a bit more. And to what extent are other relevant variables verified, such as temperature gradients, wind, humidity, density, grain size, etc?

p15 Fig6 Please describe what BT and BTL are in the caption.

p18#8 The sentence is missing a word, maybe "we" in front of "found"?

p18#20 Was it as low as 0.4?

p21#22 Please specify what is referred to as "these trees" (left branch ... left branch)

p22#15 Sparse or lacking field observations from high elevation, especially during periods with poor weather or visibility, could be another factor supporting the use of heuristics by forecasters. Another points that could be brought into the discussion is that the forecasters find it difficult to distinguish

storm slab and wind slab, and the two problems may coexist in different parts of the terrain according to how wind-affected the snow it is. Another point concerns the cornice problem, which is partly the effect of the terrain – suggesting snow and weather variables are less relevant when defining the problem.

p22#27 It could be noted that how well precipitation from NWP models agree with observations could vary significantly, according to weather situation, topography, type of models etc.

p24#5 Consider adding something like ", entry and exit" after "presence"

p24#6 Missing "as" after "such"?

p24#11 Could be other reasons than forecasting practice, e.g. lack of information, incomplete process understanding, or the fact that fading out may generally be a more gradual and slow process than the onset of problems. Here it would be interesting to include data on which forecaster produced the forecast, as individuals may enter and remove problems in different ways - or it may even differ according to day number in the forecasting period the forecaster is on duty. This could be something to consider bringing into the discussion.

References:

- Several references are missing a description of where the paper is published, please add this information:
  - Bellaire, S. and Jamieson, J. B.: On estimating avalanche danger from simulated snow profiles, 2013, 154-161.
  - Giraud, G.: MEPRA: an expert system for avalanche risk forecasting, 1992, 97-104.
  - Milbrandt, J. A., Bélair, S., Faucher, M., Vallée, M., Carrera, M. L., and Glazer, A.: The Pan-Canadian High Resolution (2.5 km) Deterministic Prediction System, Weather and Forecasting, 31, 1791-1816, 2016.
  - Müller, K., Engeset, R., Landrø, M., Humstad, T., Granan, E., and Thorset, H.: Avalanche Problem Solver (APS) - a decision support system for forecasters, 2018, 1131-1135.
  - Pozdnoukhov, A., Matasci, G., Kanevski, M., and Purves, R. S.: Spatio-temporal avalanche forecasting with Support Vector Machines, 5/1/2015, 11, 367-382, 2011.
  - Yokley, L., Hendrikx, J., Birkeland, K., Williams, K., and Leonard, T.: Role of synoptic atmospheric conditions in the formation and distribution of surface hoar, 2014 2014, 622-627.
- Some references are difficult to understand how to retrieve, please add a better description or an appropriate URL or DOI
  - European Avalanche Warning Services: Typical avalanche problems, Munich, Germany, 2017.
  - Faucher, M.: Technical note for High Resolution Deterministic Prediction System version 4.2.0, 2016.
- Please refer to the final publication, rather than the discussion paper
  - Techel, F., Ceaglio, E., Coléou, C., Mitterer, C., Morin, S., Purves, R. S., and Rastelli, F.: Spatial consistency and bias in avalanche forecasts a case study in the European Alps, Nat. Hazards Earth Syst. Sci. Discuss., 2018, 1-37, 2018.
    - https://nhess.copernicus.org/articles/18/2697/2018/

---

## Referee Comment (RC2) · Benjamin Zweifel (Referee) · 6 Oct 2020

**General comments**

A nice manuscript. When I saw the title of this paper, I was really looking forward to read it, since it promised to give insights to a decision making aid of assessing avalanche problems in operational services. It was interesting and confirmative to see, that the decision trees you derived from snow and weather data follows at least some intuitive, physical understandable rules and has some parallels to the decision aid we use in operational service, which is based on expert opinion only.

Furthermore, you nicely showed that contextual information such as presence or absence of other avalanche problems have a great influence on the appearance of an avalanche problem as well. This fact should be considered in future decision aid developments, which should – as you suggest – combine data driven approaches, as you have undertaken in your study, and expert opinions.

The next step in my eyes would be to conduct similar studies in other context, e.g. other countries, forecast services or with different definitions of avalanche problem types. Anyway, thanks for breaking the ice into this direction.

I propose hereby some minor revisions as listed below. Since I see this piece of work as very relevant I encourage the authors to undertake the suggested revisions.

**Specific comments**

Page 2, Line 2: Problems are assessed by answering four questions:….

> ➔ I'm not sure, whether the questions you mention address only avalanche problems. In my eyes, the questions describe the approach of the conceptual model of avalanche hazard assessment. Accordingly, you should say "Avalanche hazard is assessed by …."

Page 2, Line 7-10: The four references you mention here do not really refer to guidelines for applying avalanche problems. Klassen, 2014 gives a very general and qualitative description of the problem types, Lazar, 2012 shows how danger ratings patterns on specific avalanche problem types, Müller, 2016 and 2018 describes a conceptual and an operational approach of avalanche hazard assessment. None, of these studies directly addresses the assessment of avalanche problems. Therefore, I suggest to reformulate this part. By the way, up to my knowledge, no direct decision making aids or guidelines for applying avalanche problems are published so far. At SLF, we have an internal guideline (see below), which is, however, not published and I guess there are more internal guidelines existing.

[Figure]

Page 3, Line 22: Why do you exclude early and late season? They would probably be interesting for wet snow and gliding snow avalanche problems? This needs more explanation. Probably, you have to adapt on page 4, line 3 as well.

Page 12, Line 7ff: In the decision tree for storm slabs, TSS is appearing, which is a bit surprising for me. Do you have any explanations of this? You did not mention in the text.

Page 19, Figure 7: In the bars at the lower end of the figure one cannot always read the "Surf", "Act" and "Dorm" notations. Maybe, they are better described in a legend.

References: please complete the information; many references are not clear where they were published (see comment of Rune Engeset for more detail)

Page 31-34, Appendix C: The figures are to small to read, make them bigger or increase resolution.

---

## Author Comment (AC1) · 28 Oct 2020

*A nice manuscript. When I saw the title of this paper, I was really looking forward to read it, since it promised to give insights to a decision making aid of assessing avalanche problems in operational services. It was interesting and confirmative to see, that the decision trees you derived from snow and weather data follows at least some intuitive, physical understandable rules and has some parallels to the decision aid we use in operational service, which is based on expert opinion only.*

*Furthermore, you nicely showed that contextual information such as presence or absence of other avalanche problems have a great influence on the appearance of an*

*avalanche problem as well. This fact should be considered in future decision aid developments, which should – as you suggest – combine data driven approaches, as you have undertaken in your study, and expert opinions.*

*The next step in my eyes would be to conduct similar studies in other context, e.g. other countries, forecast services or with different definitions of avalanche problem types. Anyway, thanks for breaking the ice into this direction.*

*I propose hereby some minor revisions as listed below. Since I see this piece of work as very relevant I encourage the authors to undertake the suggested revisions.*

**Thank you for the positive review. We would be excited to see similar analysis done in other regions and contexts so we can learn difference perspectives of how avalanche problems are used and work towards improvements that make them more consistent, accurate, and understandable.**

*Specific comments*

*Page 2, Line 2: "Problems are assessed by answering four questions:..." I'm not sure, whether the questions you mention address only avalanche problems. In my eyes, the questions describe the approach of the conceptual model of avalanche hazard assessment. Accordingly, you should say "Avalanche hazard is assessed by ...."*

**Done.**

*Page 2, Line 7-10: The four references you mention here do not really refer to guidelines for applying avalanche problems. Klassen, 2014 gives a very general and qualitative description of the problem types, Lazar, 2012 shows how danger ratings patterns on specific avalanche problem types, Müller, 2016 and 2018 describes a conceptual and an operational approach of avalanche hazard assessment. None, of these studies*

*directly addresses the assessment of avalanche problems. Therefore, I suggest to re-formulate this part. By the way, up to my knowledge, no direct decision making aids or guidelines for applying avalanche problems are published so far. At SLF, we have an internal guideline (see below), which is, however, not published and I guess there are more internal guidelines existing.*

**Thank you for sharing this internal guideline, it is very interesting to see the structure of these simple expert-based rules as well as the footnotes that high-light special consideration (e.g. specific rules about when to use both new snow and wind slab).**

**We have updated our discussion about these existing papers and made it clearer that there is no published guidelines for assessing avalanche problems.**

*Page 3, Line 22: Why do you exclude early and late season? They would probably be interesting for wet snow and gliding snow avalanche problems? This needs more explanation. Probably, you have to adapt on page 4, line 3 as well.*

**The primary reason for omitting these time periods is limitations in our data sets (there are few hazard assessments prior to 1 December and we are missing NWP forcings for some of the spring months). We provide a better explanation of this limitation in our methods and mention in the Discussion section that studying spring conditions would be an interesting addition.**

*Page 12, Line 7ff: In the decision tree for storm slabs, TSS is appearing, which is a bit surprising for me. Do you have any explanations of this? You did not mention in the text.*

**We missed this description in the text and added a brief explanation of this split when discussing the storm slab tree. In general temperature related vari-**

**ables were a common distinguisher between storm and wind slab problems, with
storm slabs more common at warmer temperatures (air temperature was also the
top split in the wind slab tree with only weather and snowpack variables). In this
specific case, the TSS split is beneath a HN72 > 16 cm split, so the main rea-
son for this split could be to distinguish cases when the forecaster may have
assessed a dry loose problem instead of a storm slab problem because there
was lots of low density snow. We could have provided this level of detailed inter-
pretation for every split for every tree, but decided such detailed interpretations
were not needed to support our main arguments.**

*Page 19, Figure 7: In the bars at the lower end of the figure one cannot always read
the "Surf", "Act" and "Dorm" notations. Maybe, they are better described in a legend.*

**We cleaned up the overlapping labels and corrected some other hard to read text
in our figures.**

*References: please complete the information; many references are not clear where
they were published (see comment of Rune Engeset for more detail)*

**We updated our references to complete their publication locations and DOI/URL
details.**

*Page 31-34, Appendix C: The figures are to small to read, make them bigger or in-
crease resolution.*

**We increased the size of the appendix figures.**
2020-274, 2020.

---

## Author Comment (AC2) · 28 Oct 2020

*General comments*

*I really enjoyed reading this manuscript. It provided scientific value, by analyzing forecast and snow/weather model data in a thorough and novel manner with the view of providing applied value to avalanche risk forecasting and management. It applies decision tree analysis to avalanche problems, based on simulated snow and weather properties. The data are abundant, documented, structured and relevant, the analysis and methods are well described and pertinent to the problem, and the focus on the avalanche problems adds significant new knowledge to the research and operational*

[Figure]

*communities. The presentation of the results is adequate and the discussion clear and to the point. In general, the figures and tables are easy to read. It could add value to discuss the results in a more generalized context, please consider if any of these points are relevant based on your data and analysis: How may the results be transferable to other warning services and other climatic settings? How would the results be if the four (five) EAWS problems were used instead of the more detailed North American types? What about using the forecaster name/ID and the day number in his/her forecasting period as variables, was this tested?*

*I recommend publication of the paper, after minor revision addressing the detailed comments below.*

**Thank you for the positive feedback. We agree discussing the results in a broader context would add value to the manuscript and have added some discussion about the suggested points. However, we also encourage similar studies be done in other regions and forecasting contexts to properly understand the application of avalanche problems more broadly. We discuss some of these points based on our experiences working with our datasets, but are hesitant to make claims about how our results transfer to other contexts.**

**We expanded our Discussion section to acknowledge the main questions you raise and where appropriate offer some insights on how our results may transfer to different contexts. Below we answer your questions more directly:**

- **Other regions/climates: Our study area of Glacier National Park has a transitional snow climate and thus the most common avalanche problem types are storm slab, wind slab, and persistent slab (Shandro et al., 2019). Our results are likely most relevant for regions with similar climates. We assume each problem type would have similar influential weather and snowpack variables in different regions, but the threshold values may be different. Our study does not include an in-depth analysis of deep persistent slab prob-**

lems common in continental climates or wet slab/loose problems common in spring, and thus additional studies would be needed to understand their main influences.

- Other agencies: Recent studies show inconsistencies in the application of danger ratings between different forecasting agencies, and it would be fair to assume the application of avalanche problems would follow a similar pattern.

- European problem types: Analyzing the five EAWS problems would likely find similar influential weather and snowpack factors for the analogous North American problems, however the contextual factors could likely be quite unique since different agencies interpret and apply them differently. This further supports our recommendation of using this type of data-driven approach to facilitate expert discussions about current practices.

- Individual forecasters: Our analysis did not include information about the individual forecasters, but based on similar studies these could certainly be meaningful influences (e.g. Statham et al., 2018). The number of days is an interesting variable as it is likely forecasters may not be confident to change avalanche problems early on their shift. We think these would be interesting variables to include in internal analysis for specific operations with the goal of understanding inconsistencies, but the focus of this study was to highlight some of the more general contextual variables that influence avalanche problem assessments.

*Detailed comments*

*p235 In In the introduction, it could be mentioned that the utility of snow model data in operational forecasting is to a certain extent hampered by the quality and representa-*

*tiveness of input weather data. If a weather model or weather station fails to catch a precipitation event, this error will propagate to the snow model simulation and it may linger on for a long period.*

**We added a sentence to the Introduction to discuss additional limitations of snowpack models. The quality of weather inputs in particular is a pertinent issue that impacts the results of our analysis, so we agree it's worth mentioning early.**

*p322 Explain why this period (winter months with dry snow avalanche as primary concern) is chosen. Consider adding more details about the study area, e.g. latitude, tree line altitude and elevation interval or hypsometry.*

**The primary reason for our study period is limitations of our data set (there are few hazard assessments prior to 1 December and we are missing NWP forcings for some of the spring months). We now explain this in our Methods. We also added more details about the study area location.**

*p316 Consider adding "and time period" to the sub-title*

**Done.**

*p4Fig1 The sentence "Grid points used before the 2017 model update are shown with circles and grid points used after the update are shown with squares." in the caption makes limited sense to the reader at this stage in the text. Could you explain better?*

**We revised to caption to avoid reference to the model update.**

*p1125 Could you add a summary of selected variables? It may improve the reader' ability to catch up in the text.*

We added some more details about the number of variables of each type (weather, snowpack, contextual) selected for the analysis. We believe a complete list of the variables is too long for the main body of the text (94 and 56 variables for the two datasets, respectively), and is better suited for the appendix. We added some more textual summaries to help the reader follow the variable selection method and the resulting dataset used for the decision trees.

*p124-5 It is not only precipitation controlling weak layer and slab formation events, thus this sentence could be moderated or elaborated a bit more. And to what extent are other relevant variables verified, such as temperature gradients, wind, humidity, density, grain size, etc?*

Our analysis only verifies the snow height and daily snowfall amounts and does not consider other weather or snowpack variables. This would be a much more detailed analysis with non-trivial methods. The goal of our verification was to highlight that models can capture some of the dominant seasonal snowpack processes, but are also prone to errors on a case-by-case basis. We agree our explanation of the impacts on weak layer and slab formation could be explained better. The crux of our argument is the sensitivity analysis of Richter et al. (2020) who show weak layer formation in SNOWPACK is most sensitive to precipitation and air temperature inputs and slab properties are most sensitive to precipitation inputs. NWP models generally have greater uncertainties in precipitation than air temperature, hence the timing of precipitation is likely the main source of uncertainty when it comes to capturing weak layer formation with snow cover models forced with NWP data. We have revised our explanation of what our verification says about weak layer and slab formation.

*p15 Fig6 Please describe what BT and BTL are in the caption.*

**Description added to this caption (and other captions that were missing abbreviation descriptions).**

*p188 The sentence is missing a word, maybe "we" in front of "found"?*

**We improved the wording of that sentence.**

*p1820 Was it as low as 0.4?*

**Low wind speed thresholds in our trees are an artifact of the HRDPS 10 m wind speeds being lower than typical values observed in the field. This is an example of our Discussion in 4.2 about the "types of variables" being more important than the specific thresholds and that knowledge of specific model setups are needed to interpret the values in these trees.**

*p2122 Please specify what is referred to as "these trees" (left branch . . . left branch)*

**added "for surface problem types"**

*p2215 Sparse or lacking field observations from high elevation, especially during periods with poor weather or visibility, could be another factor supporting the use of heuristics by forecasters. Another points that could be brought into the discussion is that the forecasters find it difficult to distinguish storm slab and wind slab, and the two problems may coexist in different parts of the terrain according to how wind-affected the snow it is. Another point concerns the cornice problem, which is partly the effect of the terrain – suggesting snow and weather variables are less relevant when defining the problem.*

**We appreciate the additional discussion points and have added these ideas to Sect. 4.1.**

*p2227 It could be noted that how well precipitation from NWP models agree with observations could vary significantly, according to weather situation, topography, type of models etc.*

**Added a comment to the effect.**

*p245 Consider adding something like ", entry and exit" after "presence"*

**Done.**

*p246 Missing "as" after "such"?*

**Done.**

*p2411 Could be other reasons than forecasting practice, e.g. lack of information, incomplete process understanding, or the fact that fading out may generally be a more gradual and slow process than the onset of problems. Here it would be interesting to include data on which forecaster produced the forecast, as individuals may enter and remove problems in different ways - or it may even differ according to day number in the forecasting period the forecaster is on duty. This could be something to consider bringing into the discussion.*

**We added a phrase about some of these extra considerations for removing problems, and added some notes about the potential role of individual forecasters to the Discussion.**

*References*

*Several references are missing a description of where the paper is published, please add this information*

*Some references are difficult to understand how to retrieve, please add a better description or an appropriate URL or DOI*

**We updated our references to complete their publication location on DOI/URL details.**

*Please refer to the final publication, rather than the discussion paper* **Thank you for the updated citation.**

**References**

Richter, B., van Herwijnen, A., Rotach, M., and Schweizer, J.: Sensitivity of modeled snow stability data to meteorological input uncertainty, Nat. Hazard Earth Syst. Sci., 2020, 1–25, https://doi.org/10.5194/nhess-2019-433, 2020.

Shandro, B. and Haegeli, P.: Characterizing the nature and variability of avalanche hazard in western Canada, Nat. Hazards Earth Syst. Sci. 15 18, 1141–1158, https://doi.org/10.5194/nhess-18-1141-2018, 2018.

Statham, G., Holeczi, S., and Shandro, B.: Consistency and accuracy of public avalanche forecasts in western Canada, in: Proc. Int. Snow Sci. Workshop, Innsbruck, Austria, 7-12 Oct., pp. 1492–1495, http://arc.lib.montana.edu/snow-science/item/2806, 2018.